# An Improved Method for Assessing Simple Sequence Repeat (SSR) Variation in *Echinochloa crus-galli* (L.) P. Beauv (Barnyardgrass)

**Carlo Maria Cusaro, Carolina Grazioli, Francesco Zambuto, Enrica Capelli and Maura Brusoni \***

Department of Earth and Environmental Sciences, University of Pavia, Via S. Epifanio 14, 27100 Pavia, Italy; carlomaria.cusaro01@universitadipavia.it (C.M.C.); carolina.grazioli01@universitadipavia.it (C.G.); francesco.zambuto01@universitadipavia.it (F.Z.); enrica.capelli@unipv.it (E.C.)
\* Correspondence: maura.brusoni@unipv.it

**Abstract:** *Echinochloa crus-galli* (L.) P. Beauv. (barnyardgrass) is one of the most noxious weeds infesting Italian rice fields. It is characterized by high genetic intraspecific variability and has developed resistance to several classes of herbicides. The aim of our study was to assess, for the first time in Italy, the genetic diversity in *E. crus-galli* from differently managed rice fields in the Lombardy region (Northern Italy) using eight specific SSR markers. To this purpose, an amplification protocol was optimized, testing different DNA concentrations, PCR mixtures, and temperatures. A total of 48 alleles were identified in 144 samples. SSR fingerprint analysis using R 3.6.3 software (poppr, polysat, and StAMPP) allowed us to handle SSRs as codominant and polyploid data. The results suggested that genetic richness and diversity were high. The analysis of molecular variance (AMOVA) indicated that genetic variation exists mainly between agronomic managements (47.23%) and among populations (37.01%). Hierarchical clustering and PCoA were in concordance with the identification of four distinct genetic groups. Our results confirm that SSR markers represent a valuable and affordable tool for the assessment of *E. crus-galli* genetic diversity and would grant useful information to plan more targeted, effective, and sustainable control strategies against barnyardgrass. The improved methodology applied here allowed us to assess the genetic variability of an allo-hexaploid species without information loss and biased results.

**Keywords:** *Echinochloa crus-galli*; SSR markers; PCR optimization; polyploid data; genetic intraspecific variability

## 1. Introduction

Genetic variability plays a fundamental role in the adaptive response of organisms to varied environmental conditions. Weeds have evolved a genetic and phenetic plasticity that allows them to colonize very different ecosystems and to survive under the most adverse ecological stresses [1,2]. Weed intraspecific biodiversity assessment is fundamental for good weed control, including problematic cases of herbicide resistance [1,3]. Agricultural management systems influence weed biodiversity. The repeated use of herbicides that have the same mechanism of action favors the rapid evolution of herbicide resistance in weeds, aided by European regulatory constraints on the use of plant protection products (Reg. EC/1107/2009) and the practice of monoculture.

The genus *Echinochloa* (P.) Beauv. (*Poaceae*) is one of the most widespread weeds and is composed of around 50 species, mostly located in minor cereal fields and rice paddies, and therefore represents a major agricultural and economic problem [4]. In the Italian rice farming regions, the most commonly found species of this genus are *Echinochloa colonum* (L.) Link, *Echinochloa crus-galli* (L.) P. Beauv., *Echinochloa crus-pavonis* (Kunth) Schult., *Echinochloa erecta* (Pollacci) Pignatti, *Echinochloa oryzicola* (Vasinger) Vasinger, *Echinochloa hostii* (M. Bieb.) Link, and *Echinochloa phyllopogon* (Stapf) Stapf ex Kossenko [4,5].

*E. crus-galli* (barnyardgrass) is an allo-hexaploid (2n = 6x = 54) difficult to control annual weed with worldwide distribution [6–8]. It represents one of the most problematic weed species in rice fields due to its competitive abilities such as mimicking rice, exhibiting rapid germination and growth, and producing seeds in high abundance [9].

*E. crus-galli* is characterized by high genetic variability and intraspecific polymorphism [4,9–13] making the morphological identification of this weed very difficult. In addition, the different *E. crus-galli* biotypes exhibit differential herbicide susceptibility as a result of high genetic variability, allowing herbicide resistance to develop. Several studies have also demonstrated that agricultural managements, including herbicide application, could affect the genetic variability and adaptability of many species within the genus *Echinochloa*, including *E. crus-galli* [14–18]. Therefore, the assessment of genetic intraspecific diversity is very important and useful and may provide valuable information for the improvement of agricultural management practices, with particular regard to the *E. crus-galli* species. Hence, there is a need for a reproducible, rapid, and affordable methodology to analyze such variability [4,9,18–20].

For this task there are many molecular markers available to detect genetic polymorphisms of orphan plants (i.e., organisms without a publicly available reference genome sequence), such as weeds [21–27]. The majority of studies have been carried out using RAPDs and AFLPs, although these markers are not reliable due to poor reproducibility. Moreover, dominant markers may become problematic and lead to a significant loss of information when applied to polyploid species [4,9,18–20,28].

Microsatellites, or Simple Sequence Repeats (SSRs), appear to be the most appropriate markers to study genetic variability as they are highly informative and powerful tools for plant genetic analysis, being codominant, multiallelic, highly mutable, and polymorphic [29–31]. Since SSR alleles differ in length by many base pairs, SSR markers are well resolved on agarose gel [25,26]. SSRs are often recorded and analyzed as dominant markers [32–38], but this leads to loss of information about allelic variance and the presence of heterozygosity, as they are codominant [39]. However, their scoring as codominant markers in polyploid species (such as *E. crus-galli*) presents several challenges, as almost all population genetics software has been developed for haploid and/or diploid genotype analysis [40].

Only a few studies have been conducted to assess the genetic diversity of *E. crus-galli* using SSRs, due to the limited number of SSRs that have been developed specifically for this species. In fact, Lee et al. (2016) studied the genetic diversity of *Echinochloa* spp. with SSR markers from related species (*Poaceae*) [25]. Recently, new SSR markers have been identified in *E. crus-galli* through Restriction Site-Associated DNA (RAD) sequencing [22] and New Generation Sequencing (NGS) technology (Illumina) [41].

In this study, we aimed to assess the genetic diversity in *E. crus-galli* from differently managed rice fields in the Lombardy region (Northern Italy) using SSR markers developed by Chen et al. (2017) [22], scoring and analyzing them as codominant. The improved methodology applied here allowed us to assess the genetic variability of this allo-hexaploid species, without incurring a loss of information. To obtain a highly reliable, reproducible, rapid, and affordable methodology, it was necessary to optimize the whole analytic procedure. To the best of our knowledge, this study is the first to use specific SSR markers and to score them as codominant in the assessment of the genetic variability of Italian barnyardgrass populations.

Intraspecific variability assessment provides information on the ecological tolerance and competitive ability of *E. crus-galli*, which is useful for establishing effective and sustainable weed management strategies, especially in rice fields where barnyardgrass herbicide-resistant populations can cause serious problems.

## 2. Materials and Methods

### 2.1. Sample Collection

Samples were collected from 39 rice fields in the Lombardy region of Northern Italy, where a high frequency of herbicide resistance has been reported. Rice fields were managed using two different rice farming practices: conventional or pre-emergent weed control. In each paddy, sample collection was carried out in a 3 m × 6 m experimental parcel (plot) (Figure 1). We collected the maximum number of samples present (4/5 specimens) within each experimental parcel, with a total of 150 samples. Paddies with less than 4 samples per plot were excluded from the final analysis. This reduced the total number of samples to 146 collected from a total of 36 plots (Table S1).

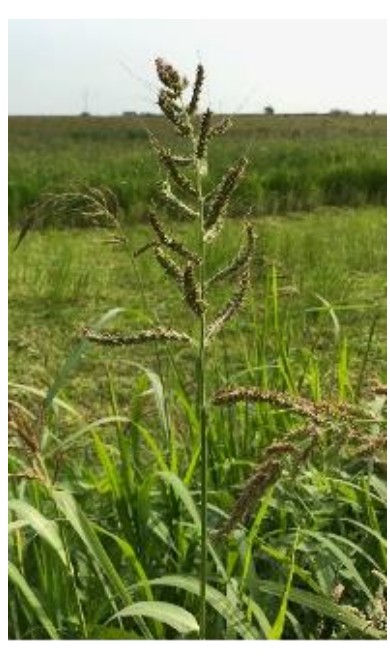

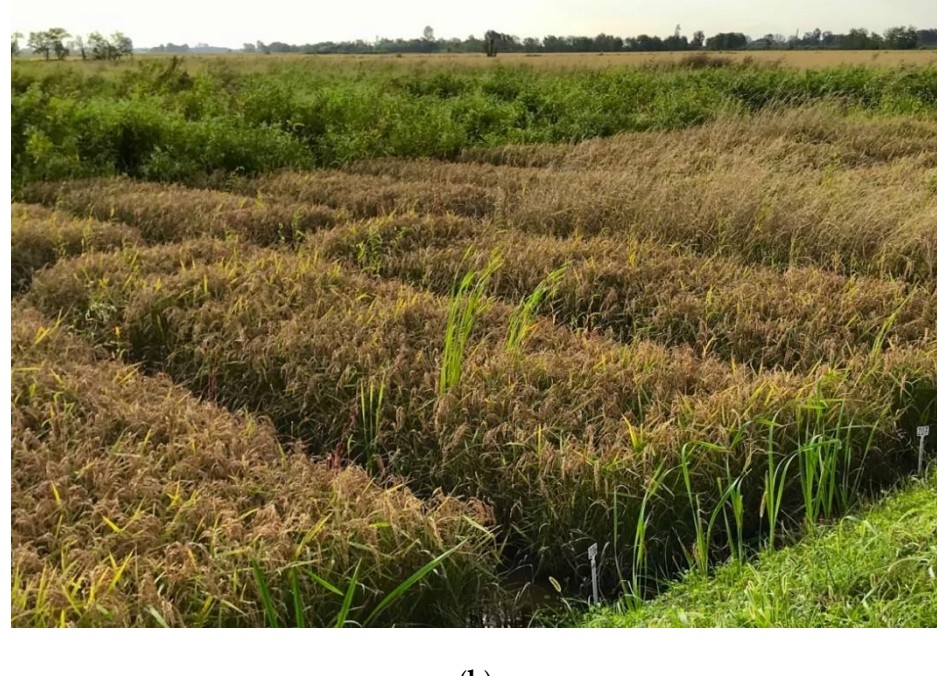

(**a**)                                                     (**b**)

**Figure 1.** Picture of: (**a**) *Echinochloa crus-galli* panicle; (**b**) experimental parcel with *Echinochloa crus-galli*.

Leaf material was stored at −5 °C until DNA extraction.

### 2.2. DNA Extraction and Quality Analysis

DNA was extracted using the DNeasy Plant Kit (QIAGEN spa, Hilden, Germany) protocol. Tissues were previously crushed using 2% CTAB buffer [42].

DNA concentration and purity were checked using Nanodrop (ThermoFisher s.p.a., Waltham, MA, USA). The absorbance ratio of the extracted genomic DNA at 260/280 nm ranged from 1.34 to 2.01, while at 230/260 nm it ranged from 0.70 to 2.27. DNA concentrations ranged from 50 ng/μL to 163 ng/μL. The quality of DNA was observed by running 4 μL of crude extracted DNA in 0.8% agarose gel. The DNA samples giving smear in the gel were re-extracted.

### 2.3. Molecular Characterization of Species

Species identification was carried out using PCR-RFLP methodology. Chloroplast DNA (cpDNA) intergenic spacer region between trnT and trnL genes was amplified with primers trn-a and trn-b1 and digested with endonuclease EcoRI (G*AATTC), whereas the entire intron region of trnL was amplified with primers trn-c and trn-d and digested with endonucleases AluI (AG*CT) and DraI (TTT*AAA) according to Amaro-Blanco et al. (2021) [43] (Table S2). The reaction–restriction mixtures were incubated overnight at 37 °C. The digested products were separated on a 2% agarose gel in 1 X TBE buffer stained with

ethidium bromide and visualized under UV rays with Molecular Imager® Gel DocTM XR + (BIO-RAD, Hercules, CA, USA). The digested product size was determined by making a comparison with a 100 bp DNA Ladder (Promega, Madison, WI, USA).

### 2.4. SSR Loci Amplification and Protocol Optimization

The SSR amplification was conducted according to the protocol described by Chen et al. (2017). The names of the SSR loci identified by Chen et al. (2017) with the corresponding repeated motifs and the primer sequences are listed in Table 1 [22]. In order to obtain reproducible results, PCR conditions (DNA template and reagent concentrations/temperatures of denaturation and annealing steps) were investigated. A series of gradient PCRs was performed to determine the most effective annealing temperatures of primers. Chen et al.'s (2017) protocol (a) and our optimized protocol (b) are compared in Table 2.

**Table 1.** *Echinochloa crus-galli* Simple Sequence Repeats. Locus name, motif, and primer sequences.

| Locus Name | SSR Motif | Primer Sequences (5′—3′) |
|:---:|:---:|:---:|
| EG1 | (TG)7 | F: GCTCCTGAACTGTGTACATTCTTGC<br>R: TCGATTCACCCTTCAGCTTCTC |
| EG2 | (TA)6 | F: CATCGGATTCAGATTGAAAGGG<br>R: GGTCGTAGGTCTATAGTCCGTAGAGTCA |
| EG301 | (AT)5 | F: GCGTCGTCAAGTCGTTCTTCTA<br>R: TGTATTCAGCTGTCGTGCATGT |
| EG302 | (ATTT)8 | F: ATTCGAACACCCATCAACCAAC<br>R: GAAACAGAAGGGAGGTGTGCTG |
| EG305 | (AT)4 | F: AGCCGTTCCTCTAGTCGGATTTCT<br>R: TATTCAGCTGCCGTGCATGTAGTA |
| EG306 | (CT)8 | F: TAAAACAAAACGACCGGCGTAA<br>R: TCAATCATTTCAGCCTTCGGAT |
| EG307 | (ATC)11 | F: AACATTGTCATCACAAATATCATCATCA<br>R: AATCAAGGAAGCCCCTTCACTC |
| EG320 | (TA)5 | F: CAACTCATAAGACAATTCAAAGGGTTT<br>R: GCATCATTTAAGCATCAAAATGACA |

**Table 2.** Protocol comparison.

| (a)   Chen et al.'s (2017) Protocol | (b)   Optimized Protocol |
|:---:|:---:|
| **PCR Mixture (in a Total Volume = 10 μL)** | **PCR Mixture (in a Total Volume = 10 μL)** |
| 0.2 μL of crude DNA (6–8 ng) extract | 2 μL of diluted DNA from crude extract (10 ng/μL) |
| 0.4 μL of each primer (0.4 μM) | 1 μL of each primer (10 μM) |
| 5 μL of Taq polymerase Ready Mix (0.27 UI) (Dongsheng Biotech) | 5.3 μL of Taq polymerase Ready Mix (0.4 UI) KAPA 2X Taq Extra Hot Start Ready-mix PCR Kit (Resnova S.r.l.) |
| (MgCl$_2$ total concentration = 1.6 mM) | Addition of 0.5 μL of MgCl$_2$ (MgCl$_2$ total concentration = 2.5 mM) |
| nuclease-free H$_2$O—ad volume | nuclease-free H$_2$O—ad volume |
| **PCR program** | **PCR program** |
| initial denaturation step at 94 °C for 4 min<br>35 cycles of:<br>94 °C for 30 s<br>relative annealing temperatures for 30 s<br>72°C for 1 min<br>final extension step at 72 °C for 10 min | initial denaturation step at 95 °C for 5 min<br>35 cycles of:<br>95 °C for 30 s<br>relative annealing temperatures for 30 s<br>72°C for 1 min<br>final extension step at 72 °C for 10 min |

DNA amplification was performed with KAPA 2X Taq Extra Hot Start Ready-mix PCR Kit (Resnova S.r.l. Genzano, Roma, Italy) through a T100 Thermal Cycler (BIO-RAD, Hercules, CA, USA). Optimization of the SSR amplification protocol was performed on a small number of the samples collected and tested in triplicate. Once improved, the protocol was extended to the analysis of all samples collected.

*2.5. DNA Fingerprinting Analysis*

The total volume of PCR amplicons was loaded on 2% agarose gel in 1X TBE buffer, stained with ethidium bromide, and separated at 100 V for 60 min. Molecular markers were visualized with Molecular Imager® Gel DocTM XR + (BIO-RAD, Hercules, CA, USA). Amplicon size was determined by making a comparison with an E-Gel® 1 Kb Plus DNA Ladder (ThermoFisher s.p.a.). For each SSR primer pair, amplicons of the same size across different isolates were considered to be the same allele. For each SSR listed in Table 1, the reproducibility of the test was validated by three replicates in which the same experimental conditions were applied and with each replicate producing a similar result. Amplicon fragment sizes were determined using the software Molecular Imager® Gel DocTM XR + (BIO-RAD, Hercules, CA, USA). A matrix of codominant data was then constructed.

*2.6. Statistical Analysis*

The number of observed alleles per locus (Na) was computed using R 3.6.3 software (poppr 2.9.3) [44–46]. The polymorphism information content (PIC) values were calculated using the formula of Liu et al. (2011) [47]:

$$\text{PIC} = 1 - \sum_{j=1}^{n} P_{ij}^2 \qquad (1)$$

where $P_{ij}$ is the frequency of $j$th allele for $i$th locus and summation extends over $n$ alleles, scored for each SSR locus, according to Prevost et al. (1999) [48] and Tiwari et al. (2016) [49].

Genotypic richness (the number of multilocus genotypes observed per population—MLG); genotypic diversity (percentage of polymorphism detected by each population—%P; Shannon–Wiener Index of MLG diversity per population—H [50]; Stoddart and Taylor's Index of MLG diversity per population—G [51]; Simpson's Index per population—lambda [52]; Evenness index per population—E.5 [53]; expected heterozygosity or Nei's unbiased gene diversity per population—He [54]; observed heterozygosity per population—Ho) were analyzed using R 3.6.3 software (poppr 2.9.3, pegas 1.0-1) [44–46,55].

Analysis of molecular variance (AMOVA) for each hierarchical comparison (between agricultural management, among and within populations) was run with 10,000 permutations by R 3.6.3 software (poppr 2.9.3, StAMPP 1.6.3) [45,46,56]. Pairwise Fst values between populations were determined with 10,000 permutations by R 3.6.3 software (polysat 1.7-5) [57,58] and plotted as a levelplot.

Genetic similarity was calculated using Nei's unbiased genetic distance with R 3.6.3 software (poppr 2.9.3) [38]. Hierarchical clustering was performed based on Ward's method to maximize the between-cluster variance with R 3.6.3 software (dendextend 1.15.2) [59]. Genotypes were sorted by PCoA (Ward's method), showing their distributions in a scatter plot, using R 3.6.3 software (poppr 2.9.3, FactoMineR 2.4, ggplot2 3.3.5) [60,61].

## 3. Results

*3.1. Molecular Characterization of Species*

Chloroplast DNA (cpDNA) intergenic spacer region nucleotide length between trnT and trnL genes differs in *Echinochloa* spp. (481 bp) and *E. crus-galli* (449 bp). Moreover, this region has an EcoRI restriction site only in *E. crus-galli*. In this method, species identification was first performed by electrophoresis of PCR products and then validated by digestion of intergenic spacer region between trnT and trnL genes with EcoRI endonuclease. For higher

accuracy, the trnL intron region was amplified and digested with AluI and DraI endonucleases, producing the same results. Among the 146 analyzed samples, 144 specimens were identified as *E. crus-galli.*

### 3.2. SSR Protocol Optimization

Figure 2 shows EG302 fingerprints obtained from a comparison of SSR amplification protocols on a small number of samples. It is possible to observe on the left (a) the DNA fingerprints obtained with the application of Chen et al.'s (2017) protocol and on the right (b) those obtained applying our modified conditions [22]. The optimized protocol allowed us to better observe the presence of multiple allelic variants (from ~180 bp to ~240 bp) due to the different sequence lengths of the SSR locus. These results proved to be reproducible in triplicate analysis. All the samples were processed by means of the optimized methodology.

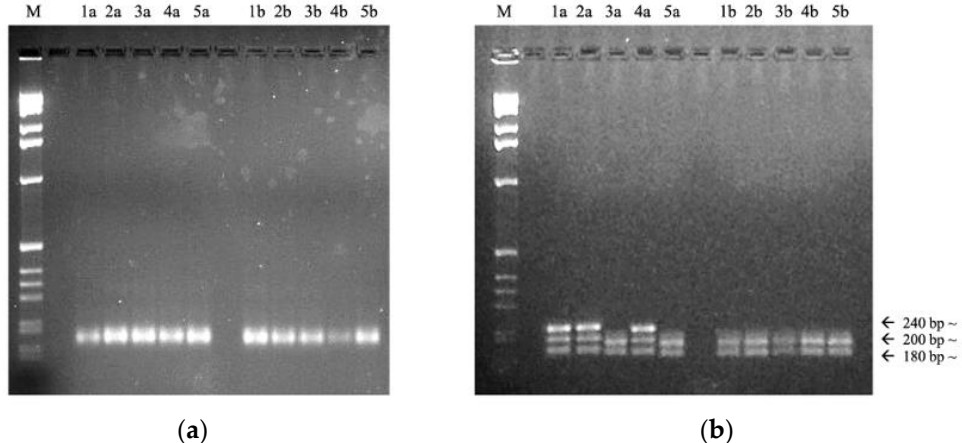

(**a**)       (**b**)

**Figure 2.** Fingerprint comparison. (**a**) Results obtained according to Chen et al. (2017) [22]. M: E-Gel® 1 Kb Plus DNA Ladder (ThermoFisher s.p.a.); 1a,2a,3a,4a,5a: different *Echinochloa crus-galli* samples from rice field "a"; 1b,2b,3b,4b,5b: different *Echinochloa crus-galli* samples from rice field "b"; (**b**) Results obtained after PCR optimization. M: E-Gel® 1 Kb Plus DNA Ladder (ThermoFisher s.p.a.); 1a,2a,3a,4a,5a: different *Echinochloa crus-galli* samples from rice field "a"; 1b,2b,3b,4b,5b: different *E. crus-galli* samples from rice field "b".

The results obtained from the different annealing temperatures tested following a gradient PCR, in comparison with the temperatures applied by Chen et al. (2017) with the same primer set [22], are reported in Table 3.

**Table 3.** Comparison between the annealing temperature (AT) of Simple Sequence Repeat markers in *Echinochloa crus-galli* according to Chen et al. (2017) and annealing temperature determined by post-gradient PCR results.

| Locus Name | AT According to Chen et al. (2017) | AT According Post-Gradient PCR Results |
|---|---|---|
| EG1 | 49 °C | 40.6 °C |
| EG2 | 51.5 °C | 50 °C |
| EG301 | 57 °C | 43.3 °C |
| EG302 | 57 °C | 48 °C |
| EG305 | 57 °C | 55 °C |
| EG306 | 57 °C | 43.2 °C |
| EG307 | 57 °C | 55.6 °C |
| EG320 | 57 °C | 46.5 °C |

### 3.3. Genetic Richness and Diversity Analysis

We analyzed 144 individuals of *E. crus-galli* from 36 paddies in the Lombardy region of Northern Italy. A total of 48 different alleles were detected using 8 SSR markers. Allele number (Na) ranged from 2 (EG307) to 12 (EG301), with an average of 6 alleles per locus. Polymorphic information content (PIC) ranged from 0.76 (EG307) to 0.98 (EG320, EG301), with an average of 0.92 per locus (Table 4).

**Table 4.** Estimates parameters of allele numbers and polymorphic information content (PIC) in the 8 pairs of Simple Sequence Repeat markers analyzed in *Echinochloa crus-galli*.

| Locus | Na | PIC |
|-------|-----|------|
| EG1   | 5   | 0.93 |
| EG2   | 3   | 0.88 |
| EG302 | 8   | 0.97 |
| EG305 | 5   | 0.96 |
| EG306 | 4   | 0.94 |
| EG307 | 2   | 0.76 |
| EG320 | 9   | 0.98 |
| EG301 | 12  | 0.98 |
| Mean  | 6   | 0.92 |

Na = average number of alleles per locus, PIC = polymorphic information content.

The analysis of the genetic richness and diversity parameters per population recorded a high diversity for the majority of plots (Table 5). The average percentage of polymorphic loci (%P) was 34.46%. The overall number of multilocus genotypes (MLG) observed was 78 and values ranged between 1 and 4. In 5 populations out of 36, an MLG value of 1 was recorded. The overall Shannon–Wiener Index of MLG diversity (H) was 29.80, with an average value of 0.82 [42]. The overall Stoddart and Taylor's Index of MLG diversity (G) was 36.64, with an average value of 2.40 [43]. The overall Simpson's Index (lambda) was 0.97, with an average value of 0.50 [44]. The overall Evenness (E.5) was 28.21, with an average value of 0.91 [45]. The overall expected heterozygosity or Nei's unbiased gene diversity (He) was 0.96, with an average value of 0.03 [46]. The overall observed heterozygosity (Ho) was 1.10, with an average value of 0.03.

### 3.4. Analysis of Molecular Variance (AMOVA)

The analysis of molecular variance (AMOVA) was carried out considering the 36 populations studied, calculating the molecular variation attributable to the differentiation between agricultural managements and among and within the populations. High percentages of variation (%V) were found between agricultural managements (%V = 47.23%) and among populations (%V = 37.01%). A lower proportion was found within populations (%V = 15.74%) (Table 6). Pairwise Fst values between populations were plotted in a levelplot and ranged between 0.000 and 0.310 (Figure S1, Table S3).

### 3.5. Hierarchical Clustering and Principal Coordinates Analysis

Hierarchical clustering identified two main genetic groups, corresponding to clusters I and II (Figure 3). Cluster I included samples from experimental parcels where only pre-emergent weed control was applied. It was divided into two subclusters (red and yellow). Cluster II comprised of samples from experimental parcels where conventional weed control was applied. It was divided into two subclusters (blue and green). Overall, four different genetic groups of accessions were identified (red, yellow, blue, green).

Likewise, Principal Coordinates Analysis identified four genetic groups (red, yellow, blue, and green) on the first three coordinates, explaining 53.67% (cumulative values) of the total variability (Figure 4).

**Table 5.** Genetic diversity parameters in *Echinochloa crus-galli*.

| Population ID | N | %P | MLG | H | G | Lambda | E.5 | He | Ho |
|---|---|---|---|---|---|---|---|---|---|
| EcgP01 | 4 | 41.15 | 3 | 1.04 | 2.67 | 0.63 | 0.91 | 0.33 | 0.67 |
| EcgP02 | 4 | 39.06 | 2 | 0.56 | 1.60 | 0.38 | 0.79 | 0.01 | 0.01 |
| EcgP03 | 4 | 41.15 | 3 | 1.04 | 2.67 | 0.63 | 0.91 | 0.02 | 0.01 |
| EcgP04 | 4 | 39.06 | 2 | 0.56 | 1.60 | 0.38 | 0.79 | 0.01 | 0.01 |
| EcgP05 | 4 | 36.46 | 2 | 0.69 | 2.00 | 0.50 | 1.00 | 0.01 | 0.01 |
| EcgP06 | 4 | 33.33 | 1 | 0.00 | 1.00 | 0.00 | – | 0.01 | 0.00 |
| EcgP07 | 4 | 38.02 | 3 | 1.04 | 2.67 | 0.63 | 0.91 | 0.02 | 0.01 |
| EcgP08 | 4 | 38.54 | 2 | 0.56 | 1.60 | 0.38 | 0.79 | 0.01 | 0.01 |
| EcgP09 | 4 | 34.90 | 4 | 1.39 | 4.00 | 0.75 | 1.00 | 0.03 | 0.02 |
| EcgP10 | 4 | 32.29 | 4 | 1.39 | 4.00 | 0.75 | 1.00 | 0.03 | 0.02 |
| EcgP11 | 4 | 37.50 | 3 | 1.04 | 2.67 | 0.63 | 0.91 | 0.02 | 0.01 |
| EcgP12 | 4 | 33.33 | 4 | 1.39 | 4.00 | 0.75 | 1.00 | 0.03 | 0.02 |
| EcgP13 | 4 | 38.54 | 3 | 1.04 | 2.67 | 0.63 | 0.91 | 0.02 | 0.01 |
| EcgP14 | 4 | 38.54 | 4 | 1.39 | 4.00 | 0.75 | 1.00 | 0.03 | 0.02 |
| EcgP15 | 4 | 35.42 | 1 | 0.00 | 1.00 | 0.00 | – | 0.01 | 0.00 |
| EcgP16 | 4 | 35.94 | 2 | 0.56 | 1.60 | 0.38 | 0.79 | 0.01 | 0.01 |
| EcgP17 | 4 | 35.42 | 1 | 0.00 | 1.00 | 0.00 | – | 0.01 | 0.00 |
| EcgP18 | 4 | 32.81 | 2 | 0.56 | 1.60 | 0.38 | 0.79 | 0.01 | 0.01 |
| EcgP19 | 4 | 32.81 | 3 | 1.04 | 2.67 | 0.63 | 0.91 | 0.02 | 0.01 |
| EcgP20 | 4 | 33.33 | 1 | 0.00 | 1.00 | 0.00 | – | 0.01 | 0.00 |
| EcgP21 | 4 | 33.33 | 1 | 0.00 | 1.00 | 0.00 | – | 0.01 | 0.00 |
| EcgP22 | 4 | 31.77 | 2 | 0.56 | 1.60 | 0.38 | 0.79 | 0.01 | 0.01 |
| EcgP23 | 4 | 31.77 | 3 | 1.04 | 2.67 | 0.63 | 0.91 | 0.02 | 0.01 |
| EcgP24 | 4 | 36.98 | 4 | 1.39 | 4.00 | 0.75 | 1.00 | 0.03 | 0.02 |
| EcgP25 | 4 | 28.13 | 3 | 1.04 | 2.67 | 0.63 | 0.91 | 0.02 | 0.01 |
| EcgP26 | 4 | 29.17 | 2 | 0.56 | 1.60 | 0.38 | 0.79 | 0.01 | 0.01 |
| EcgP27 | 4 | 31.25 | 2 | 0.69 | 2.00 | 0.50 | 1.00 | 0.01 | 0.01 |
| EcgP28 | 4 | 28.65 | 2 | 0.56 | 1.60 | 0.38 | 0.79 | 0.01 | 0.01 |
| EcgP29 | 4 | 31.25 | 3 | 1.04 | 2.67 | 0.63 | 0.91 | 0.02 | 0.01 |
| EcgP30 | 4 | 28.13 | 2 | 0.69 | 2.00 | 0.50 | 1.00 | 0.01 | 0.01 |
| EcgP31 | 4 | 27.08 | 3 | 1.04 | 2.67 | 0.63 | 0.91 | 0.02 | 0.01 |
| EcgP32 | 4 | 30.73 | 4 | 1.39 | 4.00 | 0.75 | 1.00 | 0.03 | 0.02 |
| EcgP33 | 4 | 34.38 | 4 | 1.39 | 4.00 | 0.75 | 1.00 | 0.03 | 0.02 |
| EcgP34 | 4 | 35.94 | 3 | 1.04 | 2.67 | 0.63 | 0.91 | 0.02 | 0.01 |
| EcgP35 | 4 | 39.06 | 3 | 1.04 | 2.67 | 0.63 | 0.91 | 0.02 | 0.01 |
| EcgP36 | 4 | 35.42 | 3 | 1.04 | 2.67 | 0.63 | 0.91 | 0.02 | 0.01 |
| Total | 144 | —- | 78 | 29.8 | 36.64 | 0.97 | 28.21 | 0.96 | 1.10 |
| Mean | 4 | 34.46 | 2.61 | 0.82 | 2.40 | 0.50 | 0.91 | 0.03 | 0.03 |

N = number of individuals per population, %P = percentage of polymorphism detected in each population, MLG = number of multilocus genotypes observed per population, H = Shannon–Wiener Index of MLG diversity per population, G = Stoddart and Taylor's Index of MLG diversity per population, lambda = Simpson's Index per population, E.5 = Evenness index per population (in populations where lambda is equal to 0, E.5 values could not be scored), He = expected heterozygosity per population, Ho = observed heterozygosity per population.

**Table 6.** Analysis of molecular variance (AMOVA) based on Simple Sequence Repeats in *Echinochloa crus-galli*.

| Source | DF | SS | MS | Est. Var. | % | *p* |
|---|---|---|---|---|---|---|
| Between agricultural managements | 1 | 8.20 | 8.20 | 0.11 | 47.23% | <0.001 |
| Among populations | 34 | 18.66 | 0.54 | 0.12 | 37.01% | <0.001 |
| Within populations | 108 | 4.24 | 0.03 | 0.04 | 15.74% | <0.001 |
| Total | 143 | 31.10 | 0.21 | 0.27 | 100% | |

DF = degree of freedom, SS = sum of squares, MS = mean squares, Est. var. = estimate of variance, % = percentage of total variation, *p* = *p*-value based on 10,000 permutations.

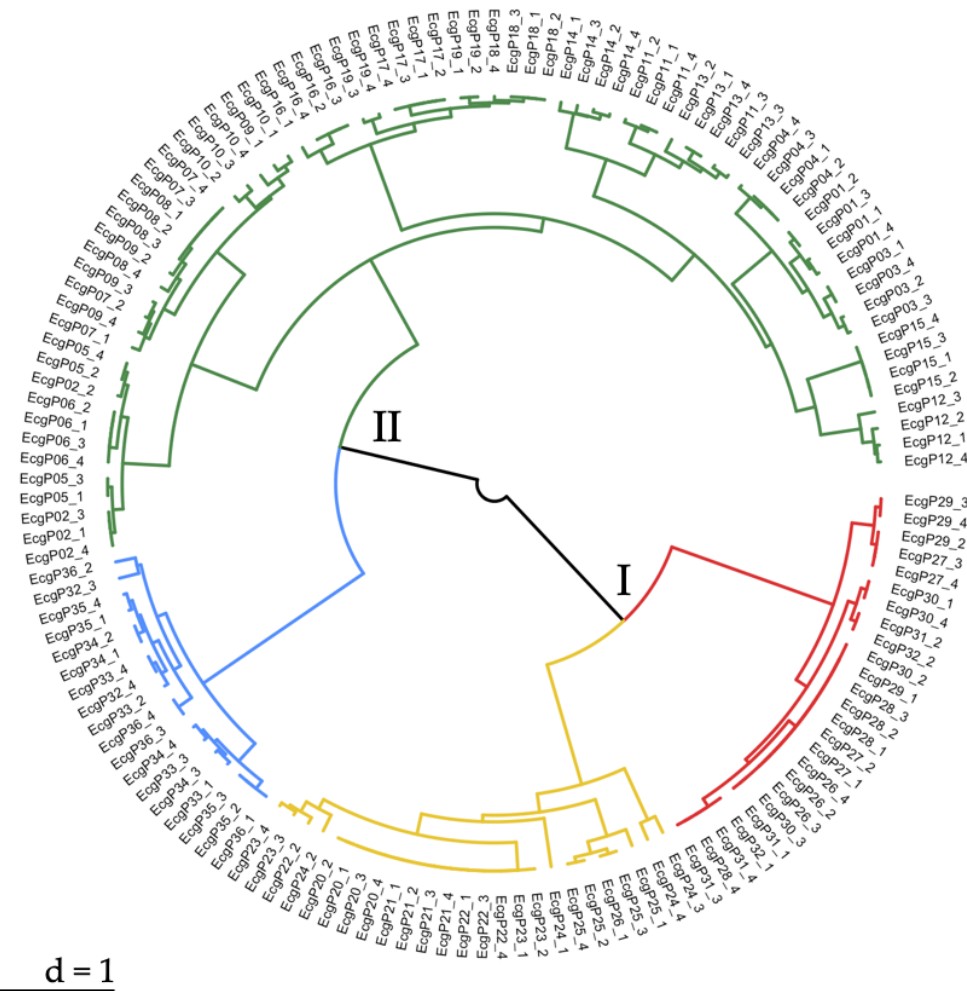

**Figure 3.** Ward.D2 hierarchical clustering based on Simple Sequence Repeats in *Echinochloa crus-galli*. *Cluster I (red and yellow)* = specimens collected from experimental parcels where only pre-emergent weed control was applied. *Cluster II (blue and green)* = specimens collected from experimental parcels where conventional weed control was applied.

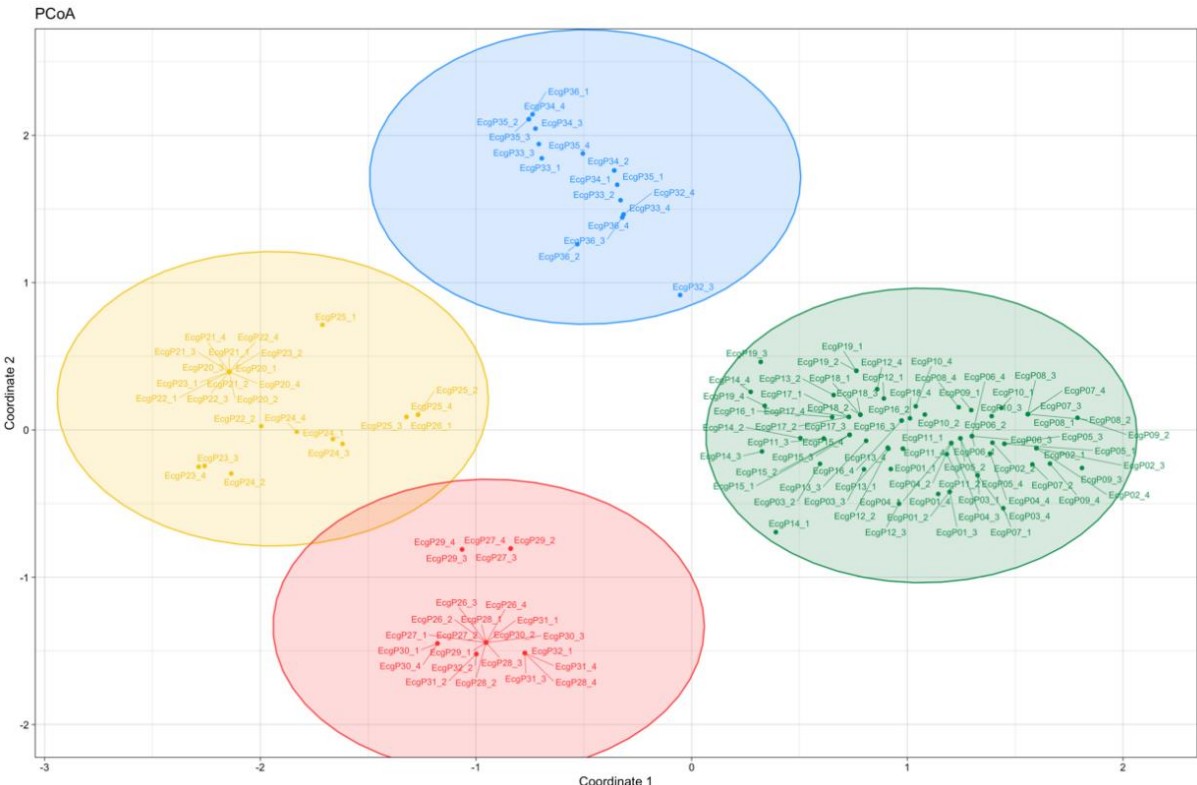

**Figure 4.** Principal Coordinates Analysis based on Simple Sequence Repeats in *Echinochloa crus-galli*. *Groups red and yellow* = specimens collected from experimental parcels where only pre-emergent weed control was applied. *Groups blue and green* = specimens collected from experimental parcels where conventional weed control was applied.

## 4. Discussion

Currently, there is little information in the literature on genetic variability studies carried out using molecular markers within the genus *Echinochloa*, and more specifically within the allo-hexaploid species *E. crus-galli*. Previously published studies had either focused on the analysis of morphological and phenological characteristics of this species or had studied genetic variability using molecular markers, which are less reliable and more subject to reproducibility problems. Only recently, some studies have approached this issue by developing and testing specific SSR markers for *Echinochloa* species, which are useful for analyzing the genetic diversity and adaptive evolution of these weeds [22,41]. Chen et al. (2017) developed specific and exclusive SSR markers for *E. crus-galli* and tested them as dominant [22], whereas Lee et al. (2019) developed cross-specific SSR markers for *Echinochloa* spp. and tested them as codominant [41]. Although this weed is particularly widespread in rice-growing areas and has, over the years, developed resistance to many classes of herbicide, no analysis of genetic variability has been conducted so far by SSR markers on Italian populations of *E. crus-galli*. In the present study, the eight SSR markers developed by Chen et al. (2017) [22] were used to assess the genetic diversity in *E. crus-galli* from rice fields in the Lombardy region (Northern Italy), scoring and analyzing them as codominant.

Molecular markers are useful tools to assay the genetic variability of plant genomes that have not been sequenced (i.e., *Echinochloa* spp.). SSRs are the most widely used molecular markers because of their high reliability, reproducibility, and affordability. However, it is often necessary to optimize and standardize the PCR mixture and profile in the procedure of DNA amplification and analysis. The dilution of the DNA template, the application of an annealing temperature lower than those applied by Chen et al. (2017), and the adjustment of Taq polymerase and MgCl$_2$ concentrations provided more successful results that were

able to obtain well-defined *E. crus-galli* SSR DNA fingerprints on agarose gel [22]. In fact, the presence of contaminants in crude DNA extract could affect the PCR outcome. Moreover, at high annealing temperatures, the PCR efficiency is reduced because only a portion of the primer molecules is able to initiate polymerization due to the high instability of their pairing with the template [23,62–65]. In addition, high Taq polymerase concentration in the PCR mixture reduces replication slippage [23,66], while a high $MgCl_2$ concentration favors more successful base pairing [67]. In the present study, the optimum Taq polymerase concentration was found to be 0.4 IU and the optimum $MgCl_2$ concentration was 2.5 mM, allowing us to obtain more defined, accurate, reproducible, and reliable DNA SSR fingerprints on agarose gel.

SSRs are commonly used in studies of population genetic diversity and structure since they are highly variable, reproducible, codominant markers for which mutational relationships between alleles can be inferred. However, their usefulness is compromised in polyploid organisms because it is difficult, or impossible, to determine allele copy number in partially heterozygous genotypes, and because inheritance patterns are complex [57,58]. To overcome this problem, many researchers have resorted to coding each allele as a dominant marker [32–38], missing part of SSRs informativeness [39,40,57,58]. In fact, it would be more appropriate to treat them as codominants [39]. For the analysis of polyploid SSR data, there are only a few computer programs available [57]. Most of the software deals with haploids and/or diploids and does not work for polyploid data. Additionally, format conversion of such data is limited [27,28,32,68]. Recently, a few statistical programs have been developed for polyploid/codominant data without information loss.

The results of this study showed that the polymorphism of SSR loci is manifested as the presence of a different number of alleles (bands on agarose gel) at each locus in the different samples. Hence, the analysis of such SSR polymorphisms was very difficult. In order to solve this problem, an extensive literature search was performed to find out useful information on the analysis of SSR markers recorded as codominant in allo-polyploid organisms. This issue was figured out using the libraries poppr 2.9.3 (Genetic Analysis of Populations with Mixed Reproduction) [45,46], polysat 1.7-5 (Tools for Polyploid Microsatellite Analysis) [57,58], and StAMPP 1.6.3 (Statistical Analysis of Mixed-Ploidy Populations) [56] implemented in the R statistical software [44]. These R packages allowed us to correctly import, read, and analyze all of our SSR hexaploid data [45,46], with particular regard to the number of alleles scored at each locus in the samples [56–58]. This allowed us to calculate parameters such as the index of genetic differentiation (Fst) and the degree of heterozygosity expected and observed in each population. Therefore, it was crucial to apply the appropriate methodology to study SSR markers as codominant for the assessment of genetic variability of the allo-hexaploid *E. crus-galli*, in order to maximize the genetic polymorphism information available.

The analysis of 144 *E. crus-galli* samples collected from 36 rice fields in the Lombardy region (Northern Italy) using the eight polymorphic SSR markers identified by Chen et al. (2017), recorded high values of genetic richness and diversity parameters per population, mostly where chemical control was applied [22]. In general, we noted that the proportion of multilocus genotypes (MLG), and thus the richness in genotypes, was higher in experimental parcels where conventional weed control was applied (MLG > 1). In contrast, we found that the proportion of MLG is commonly lower in experimental parcels where only pre-emergent weed control was applied (MLG = 1). We also found that the Shannon–Wiener (H) and Stoddart and Taylor's (G) indexes of diversity in multilocus genotypes, linked to the genotype richness, showed the same outcome, with higher values in those experimental parcels where conventional chemical control was applied. Simpson dominance index (lambda), ranging from 0 (no genotypes are different) to 1 (all genotypes are different) provides an estimate of the probability that two randomly selected genotypes are different, and it is linked to the proportion of MLG. In general, we found high values in conventional weeded experimental parcels. The Evenness index (E.5), which provides a measure of the distribution of genotype abundances, recorded a value closer to 1 in

experimental parcels with equally abundant genotypes (mostly in conventional weeded rice paddies), while a value closer to 0 was found in experimental parcels dominated by a single genotype (mostly in pre-emergent weeded rice paddies). Expected (He) and observed (Ho) heterozygosity, which are fundamental measures of genetic variation that describe the proportion of heterozygous genotypes expected under the Hardy–Weinberg equilibrium, showed high values in one population (EcgP01) [54].

According to Wright [69], if the coefficient of genetic differentiation (Fst) is less than 0.25, the level of genetic differentiation among populations is low. Our results showed different levels of genetic differentiation among *E. crus-galli* populations (Fst ranging from 0.00 to 0.31). In general, these values were high in pairwise comparisons of the weeded rice fields but were low in pre-emergent weeded rice fields. AMOVA results showed that 37.01% of the total genetic variation occurred among populations, consistently with the biology of therophytes.

Hierarchical clustering, which provides a genetic differentiation of the analyzed samples, confirmed the AMOVA results. It suggested that different agricultural practices seem to play a role in the genetic differentiation of samples into two main clusters (cluster I = experimental parcels with only pre-emergent weed control application; cluster II = experimental parcels with conventional weed control application). Clusters I and II subdivided the analyzed samples into two subclusters, based on their genetic variability. Hierarchical clustering and Principal Coordinates Analysis were in concordance with the identification of four distinct genetic groups (red, yellow, blue, and green).

The high genetic variability of *E. crus-galli* highlighted in this study, especially in conventionally weeded rice fields, might be the result of the selective pressure induced by the herbicide control. In fact, it has been reported that high levels of genetic diversity are associated with high disturbance. Genetic diversity is closely related to the adaptive capacity of a species and guarantees, both to the individual and to the progeny, the possibility to better adapt when the ecological conditions are less stable and the evolutionary pressures more intense [70].

Intensive, single-crop farming, together with the constant application of the same herbicides over time, favored the survival and development of resistant individuals and consequently caused the progressive fragmentation and local genetic differentiation of the surviving populations [17]. In any case, this fact could also be due to the biology of this species. In general, annual weed species (therophytes) that are pollinated by wind have higher levels of variation among populations [7]. Nybom (2004) showed that the genetic variability of perennial species is mostly conserved within populations, while that of annual species is mostly conserved among populations [71,72].

Such analysis could be a useful tool for preliminary screening, to obtain information on the possible risk of herbicide resistance evolution in this weed, and to predict distribution patterns of susceptible/resistant populations [41].

## 5. Conclusions

Our findings confirmed that SSR markers represent a reliable, rapid, and affordable tool to assess the genetic variability in *E. crus-galli*. The optimized protocol provided more reproducible and reliable DNA SSR fingerprints. In addition, the application of suitable software to score SSR data as codominant in polyploid species avoided biased results. High genetic intraspecific diversity was found. AMOVA revealed that there was a higher genetic diversity among (37.01%) than within (15.74%) populations. Genetic variability was found to be higher in conventional weeded paddies than in pre-emergent weeded paddies, highlighting that this weed exhibits a high adaptive capacity in response to selective pressures driven by chemical herbicide control. The results obtained from this study represent a basis for a fast-track assessment of *E. crus-galli* genetic variability that is useful for more targeted, effective, and sustainable control of this weed.

**Supplementary Materials:** The following are available online at https://www.mdpi.com/article/10.3390/d14010003/s1, Table S1: Summary of population ID and agronomic managements, Table S2: Sequences of the primers for amplification of two noncoding regions of chloroplast DNA (cpDNA), Table S3: Pairwise Fst values between populations, Figure S1: levelplot of pairwise Fst between populations.

**Author Contributions:** Conceptualization, M.B., E.C. and C.M.C.; methodology, M.B., E.C. and C.M.C.; validation, M.B. and E.C.; fieldwork, C.M.C. and C.G.; formal analysis, C.M.C. and C.G.; investigations, C.M.C., C.G. and F.Z.; resources, M.B. and E.C.; data curation, C.M.C. and C.G.; writing—original draft preparation, C.M.C. and C.G.; writing—review and editing, M.B., E.C. and F.Z.; visualization, C.M.C. and C.G.; supervision, M.B. and E.C.; project administration, M.B.; funding acquisition, M.B. All authors have read and agreed to the published version of the manuscript.

**Funding:** This research was funded by Regione Lombardia D.G. Agricoltura, Alimentazione e Sistemi verdi, bando per il finanziamento di progetti di ricerca in campo agricolo e forestale (Project number 41 EPIRESISTENZE) and by Dow AgroSciences (Corteva Agriscience since 16 June 2021). The APC was funded by Dow AgroSciences.

**Institutional Review Board Statement:** Not applicable.

**Informed Consent Statement:** Not applicable.

**Data Availability Statement:** Data are contained within the article or supplementary material.

**Acknowledgments:** We gratefully thank Fiona Jane White, Sanath Shivasuriya, Giuseppe Caporrella, Antonio Domenichetti, Angelo Fiocca, Claudio Quaroni, Daniele Rattini, Lorenzo Rampini, Tiziano Pozzi, Matteo Bartolini, and Giulia Soffiantini for technical support.

**Conflicts of Interest:** The authors declare no conflict of interest.

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
