# Peer review of "An Improved Method for Assessing Simple Sequence Repeat (SSR) Variation in Echinochloa crus-galli (L.) P. Beauv (Barnyardgrass)"

_diversity, doi:10.3390/d14010003_

Round 1
Reviewer 1 Report
Dear,
The manuscript provides a new and efficient method for the screening of diversity in Echinochloa crus-galli. However, some important aspects should to be checked and improved:
General comments:
I think the paper lacks on a table showing all the populations/samples and the local where each was collected. That table can be inserted as a supplementary file. Also, the authors should carefully read the manuscript concerning English language and standardization of many aspects (name of the species, name and acronym of the molecular marker…)
Key-words:
The name of the species already aperies in the title. It may be changed by the common name here.
Introduction:
It not common found in articles figures inserted in the introduction section, but I like of that kind of image because it helps the reader. However, I suggest to transform figure 1 and 2 in an only one figure with A and B and move the figure for material and methods or to supplementary material. In addition, also regarding the figures 1 and 2, as figure, which is an independent part of the article, their caption should be self-explanatory. First I suggest to use the name of the species in full. Second, I think is important develop the figure caption, for example "picture of the Echinocloa cruss-galli in natural/commercial fields...".
Line 56: Please change the word "so" to "in this sense". The word "so" seems informal.
Line 72: Please, after "Microsatellites" insert the acronym "(SSRs)".
Line 91: Standardize, or mention at first time the name in full of the marker "Microsatellites or Single Sequence Repeats" followed by the acronym SSR and after it mention only the acronym or use the marker name in full along the text. You should keep in mind that Microsatellites and SSRs are the same and the use of one or other in the text is confusing for the reader.
Line 94: Is the first time that NGS appears in the text, please insert the name in full.
Material and methods:
Line 120: Here is hard for me to following. The DNA extraction was performed using DNeasy Kit or CTAB method?
Lines 122-125: Keep in mind that the obtained values in the spectrophotometer are showing a kind of problem in the DNA samples, mainly the absorbance in 230/260 nm. In addition, the DNA concentrations are much low with respect to the protocol of DNA extraction. In addition, an agarose gel was performed to verify the integrity of DNA samples?
Line 133: Problem in this reference (Chen et al.,), please add the year of the publication. Check, it occurs along the manuscript.
Line 140 (Table 1): Please, table caption should be more informative, use the name of the marker in full and insert the name of the species in which these oligonucleotides were used. I suppose the oligonucleotides sequences are in 5' to 3' sense, is it true? Please insert that information.
Line 142 (Table 2): Insert also in the table the name of the DNA polymerase manufacturer. Keep in mind that, as figures, tables are independent parts of the article and should be self-explanatory.
Line 152: Standardize (BIO-RAD, 145 Hercules, California, USA).
Line 158: Standardize (BIO-RAD, 145 Hercules, California, USA).
Line 181: Insert the version of the softwares used.
Results:
Figure 3: Insert the name of the species in full and also the manufacturer of the products cited.
Line 210 (Table 3): In which species? Using which marker?
Line 221 (Table 4): In which species?
Line 238 (Table 5): Species name in full.
Line 258 (Table 6): In which species? based in which marker?
Line 272 (figure 4): In which species? based in which marker?
Line 280 (figure 5): In which species? based in which marker?
Discussion
Line 286: exclude the word “both”.
Line 289: exclude the words “in this weed”.
Line 304: insert “…the most widely molecular marker used,…”
Lines 321-323: Are you mean that the specificity of an oligo is lower in high temperatures? Please check that sentence.
Line 343: exclude the words “In fact, the”
Line 431: The reference Nybom et al., lacks the year of publication near the name.
Conclusion
As the article show results and discussion in separated section, the conclusion is expected to be more concise. Only the important finds that differ this paper from the others should be pointed.
Author Response
- I think the paper lacks on a table showing all the populations/samples and the local where each was collected. That table can be inserted as a supplementary file. A table showing a summary of sampling plot, number of samples per plot, agronomic managements and population ID has been inserted in the supplementary file material (Table S1).
- Language and names standardization. Based on your suggestion, the name of the species, the name and acronym of the molecular marker and others have been standardized.
- Key-words - The name of the species already aperies in the title. It may be changed by the common name here. We thank you for your suggestion. However, we prefer to keep the species name (Echinochloa crus-galli) here. The common name (barnyardgrass) may not be as well-known as the species name.
- Introduction - It not common found in articles figures inserted in the introduction section, but I like of that kind of image because it helps the reader. However, I suggest to transform figure 1 and 2 in an only one figure with A and B and move the figure for material and methods or to supplementary material. In addition, also regarding the figures 1 and 2, as figure, which is an independent part of the article, their caption should be self-explanatory. First I suggest to use the name of the species in full. Second, I think is important develop the figure caption, for example "picture of the Echinocloa cruss-galli in natural/commercial fields...”. We have followed your suggestion to change the layout of Figures 1 and 2 as Figure 1 (a/b). We have developed the caption of figures as “Figure 1 (a/b). Picture of: (a) Echinochloa crus-galli panicle; (b) Experimental parcel with Echinochloa crus-galli”. We have also substituted Figure 1b with a picture of an experimental parcel. Then, we have moved Figure 1 (a/b) to material and methods.
- Material and methods - Line 120: Here is hard for me to following. The DNA extraction was performed using DNeasy Kit or CTAB method? CTAB 2% buffer was used to grind plant samples. The DNeasy plant kit from QIAGEN was used for DNA extraction from crushed tissues.
- Lines 122-125: Keep in mind that the obtained values in the spectrophotometer are showing a kind of problem in the DNA samples, mainly the absorbance in 230/260 nm. In addition, the DNA concentrations are much low with respect to the protocol of DNA extraction. In addition, an agarose gel was performed to verify the integrity of DNA samples? Low A260/A230 nm ratios may be the result of carbohydrate carryover and residual phenol from nucleic acid extraction. Low A260/A230 nm ratios were recorded only in a few DNA samples that were appropriately diluted prior to amplification. Average absorbance ratios have been substituted with range values. Moreover, in some extracted samples DNA concentration is low, but enough to run PCR after dilution, as specified in the MS (Table 2). We verified the integrity of the crude extracted DNA on agarose gels. Samples giving smear were re-extracted.
- Problem in this reference (Chen et al.,), please add the year of the publication. Check, it occurs along the manuscript. All references have been checked and corrected (Chen et al. 2017)
- Line 140 (Table 1): Please, table caption should be more informative, use the name of the marker in full and insert the name of the species in which these oligonucleotides were used. I suppose the oligonucleotides sequences are in 5' to 3' sense, is it true? Please insert that information. We have implemented table caption and inserted the information into the table, as you suggested.
- Line 142 (Table 2): Insert also in the table the name of the DNA polymerase manufacturer. Keep in mind that, as figures, tables are independent parts of the article and should be self- explanatory. The name of the DNA polymerase manufacturer has been inserted into the table.
- Line 181: Insert the version of the softwares used. Software versions have been inserted.
- Discussion - Lines 321-323: Are you mean that the specificity of an oligo is lower in high temperatures? Please check that sentence. The sentence in question has been checked and rewritten so that it is more understandable to the reader. We wanted to assert that at high annealing temperatures, the PCR efficiency is reduced because only a portion of the primer molecules are able to initiate polymerization due to the high instability of their pairing with the template.
- Conclusion - As the article show results and discussion in separated section, the conclusion is expected to be more concise. Only the important finds that differ this paper from the others should be pointed. We have rewritten the conclusion of the MS according to your suggestions to make it more fluent, concise, and focused.
- Text corrections and names standardization. We have accepted all of your suggestions regarding text corrections and names standardization to be reported in the MS.
- Moderate English changes should be performed. The MS was reviewed by a native English speaker from USA, Dr. Fiona Jane White. Her revisions are visible as “Track Changes” in the MS.
Reviewer 2 Report
Manuscript has improved significantly therefore, it can be accepted for publication.
Author Response
- Manuscript has improved significantly therefore, it can be accepted for publication. The Authors thank you so much.
Reviewer 3 Report
This paper has extensively improved. If they still keep the first version, I will reject immediately this paper. However, this version still need more work before acceptable for publication. Please see details in the attached file.

Author Response
- The current manuscript is not well structured and hard to read. For example, the information shown in the introduction part is long and rambling. Please improve this part by focusing on your main objectives, such as E. crus-galli in Northern Italy, SSR methods, etc.. The MS introduction was revised according to your suggestion, to make it more fluent and focused regarding the main aims of our study.
- Are Figures 1 and 2 obtained through the process of conducting your study? Figures 1 and 2, whose layout was modified as Figure 1 (a/b) with reference to the suggestions of Reviewer 1, were captured during the sampling phase of the Echinochloa crus-galli individuals under study.
- Line 60: citation should be [14-17], line 69: citation should be [21-26], please revise for others. Citations have been corrected following your suggestions.
- The conclusion should be concise, focusing on the significant points. The conclusion part should not include citations, please mention them in the discussion part. We have rewritten the conclusion of the MS according to your suggestions to make it more fluent, concise, and focusing on the significant points. Citations have been mentioned in the discussion part.
- Moderate English changes should be performed. The MS was reviewed by a native English speaker from USA, Dr. Fiona Jane White. Her revisions are visible as “Track Changes” in the MS.
Reviewer 4 Report
I could not comprehend yet what is the role of ploidy differences of various populations on clustering. The gel pictures presented are pretty lame and do not have polymorphisms that have high heterozygosity or Fst. I highly recommend to replace these gel pictures with the ones that show of intra and interpopulation polymorphisms. Some of the complicated discussion that do not have much to do with current analysis can be removed.
Author Response
- I could not comprehend yet what is the role of ploidy differences of various populations on clustering. Echinochloa crus-galli is an allo-hexaploid species. In the assessment of genetic intraspecific variability of this weed, a variable number of alleles were identified in each sample analyzed, for each SSRs locus investigated. To properly analyze a species with this kind of ploidy, it was necessary to use specifically dedicated software (i.e. poppr, polysat, StAMPP). The clustering is the result of the analysis of allo-hexaploidy samples by such software.
- The gel pictures presented are pretty lame and do not have polymorphisms that have high heterozygosity or Fst. I highly recommend to replace these gel pictures with the ones that show of intra and interpopulation polymorphisms. Figure 2 (a/b) does not address the analysis of genetic polymorphism in Echinochloa crus-galli by SSR markers. This figure has been inserted in the section "protocol optimization" to highlight the comparison of two SSRs amplifications carried out before (a) and after (b) the optimization of the protocol itself. Optimization of the SSR amplification protocol was performed on a small number of the samples collected and tested in triplicate. Once improved, the protocol was extended to the analysis of all samples collected.
- Some of the complicated discussion that do not have much to do with current analysis can be removed. The discussion part has been shortened and made more focused on the topics under analysis.
- Moderate English changes should be performed. The MS was reviewed by a native English speaker from USA, Dr. Fiona Jane White. Her revisions are visible as “Track Changes” in the MS.
Reviewer 5 Report
The work is very interesting and concerns an important topic related to the possibilities of better weed control. The SSR method is one of the best methods used to assess genetic diversity within a population, and allows drawing conclusions about genetic changes occurring in populations. However, the work before publication needs to be improved, especially in the description of the methodology, which is in many places unclear.
in the description of the material and methods, the authors write that the samples were collected in 39 fields of rice, a total of 150 samples were collected. while in the results and discussions, the authors analyze 36 populations consisting of 144 samples in total. what are the reasons for these divergences and on what basis were the samples divided into populations?
In the results and discussions, the authors write that the samples were taken from plots with different cultivation conditions, such characterization is missing in the description of the material and methods, the authors should add such information and specify how many plots of particular types of cultivation were there and how many samples were collected from them.
DNA isolation - what method was the isolation of DNA? with DNeasy plat kit or CTAB? what does it mean when it says that the samples have been crushed following CTAB method?
lines 123-125 is a description of the results, not the methods - this information should be transferred to the results chapter.
chapter 2.3 molecular characterization of species - since the authors include such a methodology, they should describe the results for this part of the work - it will probably explain the question of 36 population and 144 samples, but these results should be slightly expanded and the results should be transferred to the chapter. the basis on which the authors made the classification to the species should be specified.
I have some doubts about the size of individual populations, is it reliable to draw conclusions based on the analysis of 4 samples? in that case, it might be better to avoid naming the samples with a population and identifying them as samples coming from a given location
Author Response
- In the description of the material and methods, the authors write that the samples were collected in 39 fields of rice, a total of 150 samples were collected, while in the results and discussions, the authors analyze 36 populations consisting of 144 samples in total. What are the reasons for these divergences and on what basis were the samples divided into populations? Sample collection section has been implemented in the MS, in order to make it better explanatory. Samples were collected from 39 rice fields. In each paddy, sample collection was carried out in a 3x6 m experimental parcel (plot). We collected the maximum number of samples present (4/5 specimens) in each experimental parcel, with a total of 150 samples. Paddies with less than 4 sample per plot were excluded from the analysis. This reduced the total number of samples to 146, collected from a total of 36 plots. Moreover, 144 specimens from 36 paddies were identified as crus-galli by PCR-RFLP methodology (Table S1).
- In the results and discussions, the authors write that the samples were taken from plots with different cultivation conditions, such characterization is missing in the description of the material and methods, the authors should add such information and specify how many plots of particular types of cultivation were there and how many samples were collected from them. In materials and methods we have specified that samples were collected from 39 rice fields managed using two different rice farming practices: conventional or pre-emergent weed control. Moreover, in Table S1 there are listed the experimental parcels (plots) where Echinochloa samples have been collected, the number of individuals collected per plot, the agronomic management applied in each plot and the population identification code.
- DNA isolation - what method was the isolation of DNA? with DNeasy plant kit or CTAB? what does it mean when it says that the samples have been crushed following CTAB method? CTAB 2% buffer was used to grind plant samples. The DNeasy plant kit from QIAGEN was used for DNA extraction from crushed tissues.
- lines 123-125 is a description of the results, not the methods - this information should be transferred to the results chapter. We have taken in account your suggestion. However, this information does not represent one of our research findings, but is to be considered as a methodological step. Therefore, we prefer to maintain it in material and methods, as in the majority of genetic diversity assessment articles.
- chapter 2.3 molecular characterization of species - since the authors include such a methodology, they should describe the results for this part of the work - it will probably explain the question of 36 population and 144 samples, but these results should be slightly expanded and the results should be transferred to the chapter. the basis on which the authors made the classification to the species should be specified. Molecular characterization of species has been improved focusing and detailing the whole methodology applied and the relative results, following your suggestions.
- I have some doubts about the size of individual populations, is it reliable to draw conclusions based on the analysis of 4 samples? in that case, it might be better to avoid naming the samples with a population and identifying them as samples coming from a given location. In each paddy, sample collection was carried out in a 3x6 m experimental parcel (plot). We collected the maximum number of samples present in each experimental parcel. Samples set collected from each experimental parcel was considered as a “population” of few individuals. Therefore, in the MS we have preferred to maintain the sample set named as “population”.
Round 2
Reviewer 3 Report
This paper is ok now, and can be accepted for publication. Thank you for your hard work in revising this paper.
Reviewer 5 Report
the authors complied with the comments and improved their work. the manuscript is now clearer and more understandable to the reader
This manuscript is a resubmission of an earlier submission. The following is a list of the peer review reports and author responses from that submission.
Round 1
Reviewer 1 Report
The authors need to make some corrections, as mentioned below.
English language corrections are required from a Native speaker
Add more information in the abstract.
Restructure Introduction include a solid hypothesis.
Replace old references with recent ones.
Add more information in the Introduction it is difficult to understand how this technology can help in the omics era.
Material and methods are not clear; please elaborate
The figures are poorly presented and difficult to interpret use high resolution if possible.
Elaborate on the discussion section with more information and recent information on similar crops.
Add a conclusion section.